# A Comparative Study of Deep Neural Networks for Real-Time Semantic Segmentation during the Transurethral Resection of Bladder Tumors

**DOI:** 10.3390/diagnostics12112849

**Published:** 2022-11-17

**Authors:** Dóra Varnyú, László Szirmay-Kalos

**Affiliations:** Department of Control Engineering and Information Technology, Budapest University of Technology and Economics, Műegyetem rkp. 3, 1111 Budapest, Hungary

**Keywords:** bladder cancer, white-light cystoscopy, semantic segmentation, convolutional neural network, transfer learning, guided filtering, unsharp masking

## Abstract

Bladder cancer is a common and often fatal disease. Papillary bladder tumors are well detectable using cystoscopic imaging, but small or flat lesions are frequently overlooked by urologists. However, detection accuracy can be improved if the images from the cystoscope are segmented in real time by a deep neural network (DNN). In this paper, we compare eight state-of-the-art DNNs for the semantic segmentation of white-light cystoscopy images: U-Net, UNet++, MA-Net, LinkNet, FPN, PAN, DeepLabv3, and DeepLabv3+. The evaluation includes per-image classification accuracy, per-pixel localization accuracy, prediction speed, and model size. Results show that the best F-score for bladder cancer (91%), the best segmentation map precision (92.91%), and the lowest size (7.93 MB) are also achieved by the PAN model, while the highest speed (6.73 ms) is obtained by DeepLabv3+. These results indicate better tumor localization accuracy than reported in previous studies. It can be concluded that deep neural networks may be extremely useful in the real-time diagnosis and therapy of bladder cancer, and among the eight investigated models, PAN shows the most promising results.

## 1. Introduction

Bladder cancer is one of the ten most common types of cancer and is also reported to be a leading cause of death in Western countries [1]. The diagnosis is most often based on white-light cystoscopy imaging, which is excellent for the detection of papillary bladder tumors, but small or flat lesions may not be visible clearly enough [2]. Furthermore, the interpretation of cystoscopy findings can vary from examiner to examiner, since no two physicians have the same skills and experience [3]. Overall, it is estimated that up to 20% of tumors are overlooked during white-light cystoscopy [4].

Deep learning (DL) methods are developing rapidly, and as a result, neural network-based bladder cancer diagnosis is becoming increasingly popular [4,5,6]. DL solutions can not only achieve diagnostic accuracy similar to that of experienced specialists [2,7], but their results are also objective and reproducible [3], as they are based on mathematical operations. Using neural networks to help diagnose bladder cancer could result in fewer tumors going undiagnosed, which could save lives. In addition, fewer healthy patients may need to undergo unnecessary biopsies, which is a process that is invasive, painful, and expensive.

One of the first studies to examine deep learning-based bladder cancer diagnosis is the work of Ikeda et al. [8], who used the GoogLeNet model [9] to classify cystoscopic images. Their network was pre-trained on the ImageNet database [10] of 1.2 million natural images. Subsequently, using the obtained weights as initial values, the model was fine-tuned to classify white-light cystoscopic images as cancerous or normal. Their training data consisted of 1336 normal and 344 tumor images, but the tumor images were augmented with random rotation and blurring to have the same number as the normal images. Their model achieved a sensitivity of 89.7% and a specificity of 94.0% on the test data, with 78 of the images being true positive, 315 true negative, 20 false positive, and 9 false negative.

Another example is the work of Ali et al. [2], who proposed an artificial intelligence diagnostic platform that operated on high-quality blue-light cystoscopy images and predicted cancer malignancy, invasiveness, and grade. In their platform, they implemented four convolutional neural networks (VGG-16 [11], ResNet-50 [12], MobileNetV2 [13] and InceptionV3 [14]), which were pre-trained and fine-tuned via transfer learning. Their dataset consisted of 216 blue-light cystoscopy images of bladder tumors classified according to the 2004 World Health Organization (WHO) classification. The prediction results of the proposed CNN models were compared with each other as well as with the ratings of two experienced urologists. Results showed that the mean sensitivity obtained by any of the CNNs was at least 15% higher than that of the two urologists. The highest mean classification sensitivity, 91.81%, was achieved by the MobileNetV2 model.

Similar results were reported by Yang et al. [7], who examined the ability of three convolutional neural networks (LeNet [15], AlexNet [16] and GoogLeNet [9]), the EasyDL deep learning platform [17], and urologist experts to classify bladder cancer on white-light cystoscopy images. The highest accuracy, 96.9%, was achieved by EasyDL, which was followed by GoogLeNet with 92.54%. Overall, the deep learning-based methods had an average accuracy of 86.36%, while urologist experts achieved only 84.09%.

Further examples of AI-assisted cystoscopic image classification can be found in the review published by Negassy et al. [18].

Although these results are promising, classification can only assist in cancer diagnosis but not its treatment, the transurethral resection, during which the precise location of the tumor is required so that it can be removed.

The accurate segmentation of bladder cancer on cystoscopic images using deep learning has been reported by Shkolyar et al. in 2019 [4]. They developed CystoNet, a CNN-based analysis platform that segments bladder tumors on white-light cystoscopy video frames with a per-frame sensitivity and specificity of 90.9%. The accuracy of tumor localization was not reported. Furthermore, their data mostly consist of papillary urothelial carcinoma, which is a characteristic, finger-like growth in the bladder lining extending into the center of the bladder, and it is thus typically well detectable even by the human eye.

Zhang et al. [19] utilized an attention mechanism-based model to segment cystoscopic images. They have introduced several attention modules into the classical U-Net model and found that it improves the tumor segmentation performance: the new network achieved a Dice score of 82.5%, while the classical U-Net was found to obtain only 67% in [5].

Lastly, Yoo et al. [6] deployed a mask region-based convolutional neural network with a ResNeXt-101-32×8d-FPN backbone to identify and grade bladder tumors on white-light and narrow-band cystoscopy images. The sensitivity and specificity they achieved were 95.0% and 93.7%, respectively, but the Dice score of localization was only 74.7%.

It can be concluded that there is still room for improvement in terms of bladder tumor localization accuracy. Since there are examples of classical deep neural networks (DNNs) achieving excellent results in other image segmentation tasks [20,21], the idea of a comprehensive comparison between different DNNs for cystoscopic image segmentation also arises. By identifying a model with good bladder tumor localization accuracy and real-time speed, the success rate of transurethral resections may be significantly improved.

In this paper, we compare eight DNNs for the semantic segmentation of white-light cystoscopy images: U-Net [22], UNet++ [23], MA-Net [24], LinkNet [25], FPN [26], PAN [27], DeepLabV3 [28] as well as DeepLabV3+ [29], and we examine the potential benefits of image pre-processing. Evaluations cover per-image classification accuracy, per-pixel localization accuracy, prediction speed, and model size.

The paper is organized as follows. Materials and methods are described in Section 2, which includes the investigated DNNs, the image pre- and post-processing pipeline, the network training, and the measurement setup. The results of the evaluation are presented in Section 3 and discussed in Section 4. Finally, the paper is concluded in Section 5.

## 2. Materials and Methods

The task is to produce a high-precision semantic segmentation map that locates bladder cancer as well as other lesions and artifacts on white-light cystoscopic images. The obtained segmentation map can be overlaid on the original image (Figure 1) to assist the transurethral resection of bladder tumors in real time. In this section, we present the DNN models to be evaluated (Section 2.1), the image processing pipeline (Section 2.2), the network training (Section 2.3), and the measurement setup that was used for the evaluations (Section 2.4).

### 2.1. The Deep Neural Networks to Be Compared

In the field of image analysis, convolutional neural networks (CNNs) are the most widely used deep learning solutions [30]. They take advantage of the spatial arrangement of images, i.e., the fact that information is usually local in images [5], so it may be sufficient to examine only a part of the image content at a time. CNNs do this by running convolutional filters over the input image, which share the learnable weights between image parts, thereby significantly reducing the number of parameters to be optimized. These networks are able to model complex hierarchical functions and train deep models efficiently [31].

An encoder–decoder architecture is a special type of model design that consists of two parts: an encoder that compresses the input into a latent space vector, and a decoder that predicts the output from this intermediate representation. Encoder–decoder CNNs are particularly suited for the task of image segmentation, as they are able to model the complex underlying features of the input image in latent space and then produce an output image of the same size as the input, i.e., a segmentation map containing the classification of each pixel. In the following, we review the encoder–decoder CNNs that we evaluate in this paper for cystoscopy image segmentation.

U-Net [22] is a convolutional neural network designed specifically for biomedical image segmentation. It gets its name from the fact that the model consists of a downsampling part (encoder) and an upsampling part (decoder), which form a U-shaped architecture. The encoder of U-Net applies successive 3×3 convolutions, each followed by ReLU activation [32] and maximum pooling. While the spatial resolution is reduced, the feature dimension of the image is simultaneously increased to aggregate high-level semantic information. The resulting latent representation is then upsampled in the decoder to obtain a precise segmentation map. To improve performance, skip connections are introduced between encoder blocks and their corresponding decoder parts. This helps to propagate information from early layers that still retain fine input details to deeper layers that lose these details due to successive downsampling.

UNet++ [23] is a new variant of the U-Net architecture. It is based on the hypothesis that when the encoder and decoder feature maps are semantically similar, the optimizer has an easier optimization problem to solve. Therefore, skip connections are redesigned to include a dense convolution block in which the number of convolutional layers depends on the pyramid level. Basically, this dense convolution block brings the semantic level of the feature maps of the encoder closer to that of the decoder.

UNet++ was reported to give better accuracy than U-Net in multiple medical image segmentation tasks, such as nuclei segmentation in microscopy images, liver segmentation in abdominal CT scans, and polyp segmentation in colonoscopy videos [23].

MA-Net (Multi-scale Attention Network) [24] is a variant of U-Net that introduces self-attention to its architecture. It uses two blocks: the Position-wise Attention Block (PAB) and the Multi-scale Fusion Attention Block (MFAB). The PAB extracts spatial dependencies between pixels by a self-attention mechanism, while the MFAB captures channel dependencies between any feature map by attention-based multi-scale semantic feature fusion. MA-Net was originally designed for liver and tumor segmentation and has been shown to perform superiorly to U-Net and UNet++ in the segmentation of liver CT scans [24].

LinkNet [25] is an encoder–decoder type CNN that aims to achieve not only good segmentation accuracy but also a small size, i.e., few learnable parameters. This is obtained by introducing residual skip connections that bypass information from the input of each encoder block to the output of its corresponding decoder block. Unlike U-Net skip connections, where layers are concatenated, here, the features are summed, which does not increase the number of input channels and thus the number of parameters needed in the following decoding blocks [33]. This results in a small and efficient neural network.

FPN (Feature Pyramid Network) [26] is a feature extractor that builds high-level semantic feature maps at multiple scales, i.e., a feature pyramid [34]. This process does not depend on the backbone CNN, so it can be considered a generic solution. The construction of the feature pyramid involves a bottom–up pathway and a top–down pathway. The bottom–up pathway is the feed-forward computation of the backbone CNN, which creates the feature map hierarchy with a scaling step of 2. The top–down pathway down-propagates the lower spatial resolution but semantically higher level information from the upper levels of the pyramid. The features of the two pathways are fused via lateral connections. When the task is image segmentation, the different levels of the feature pyramid are used for predicting masks at different scales.

PAN (Pyramid Attention Network) [27] is a convolutional neural network designed to exploit global contextual information in semantic segmentation. First, a feature pyramid is generated using ResNet-101 [12] (encoding). Then, a Feature Pyramid Attention (FPA) module is used to gather dense pixel-level attention information. FPA is a center block between the encoder and decoder that fuses features from under three different pyramid scales and combines them with global average pooling to improve performance. Finally, a Global Attention Upsample (GAU) module on each decoder layer provides high-level features as guidance for low-level information to generate the final segmentation map.

DeepLabv3 [28] is an improvement upon the DeepLabv2 semantic segmentation architecture [35] (which itself can be traced back to the original DeepLab model [36]). To address the problem of segmenting objects at multiple scales, modules are designed to perform atrous convolutions in cascade or in parallel, capturing multi-scale context using multiple atrous rates. In addition, the Atrous Spatial Pyramid Pooling (ASPP) module of DeepLabv2 has been augmented with image-level features that encode global context and increase performance. Unlike DeepLabv2, the new model no longer needs DenseCRF post-processing to segment objects precisely.

DeepLabv3+ [29] extends DeepLabv3 by a simple yet effective decoder module that improves segmentation results, especially at object boundaries. This architecture is the first to combine spatial pyramid pooling with the encoder–decoder structure. Furthermore, inspired by Xception [37], depthwise separable convolution is applied to both the ASPP module and the decoder, resulting in an even faster and higher-precision model.

### 2.2. The Image Processing Pipeline

The goal is real-time assistance in the transurethral resection of bladder tumors. The camera stream from the cystoscope is usually of poor quality, which can hinder the performance of the neural network and the operating urologist. Therefore, we execute image enhancement as a pre-processing step, which may be configured or turned off by the urologist. After the neural network predicts the segmentation map, the outline polygons of the detected objects are extracted and overlaid on the original image to draw the doctor’s attention to possible lesions. Overall, our image processing pipeline consists of three phases:*Pre-processing*: noise reduction, contrast enhancement, and data standardization.*Segmentation*: a deep neural network predicts the segmentation map.*Post-processing*: outline polygons are extracted from the segmentation map and overlaid on the original image to show doctors the detected lesions.

A flowchart of the segmentation pipeline is displayed by Figure 2.

#### 2.2.1. Pre-Processing

Cystoscopic images may be blurred and have low contrast depending on the actual cystoscopic camera and the circumstances of the image capture. To obtain a solution that is robust to these phenomena, image pre-processing is needed during the training and recognition, which prepares the input images to allow the neural network for efficient lesion detection. For this reason, noise reduction and contrast enhancement have been incorporated into the training and detection pipelines. Noise reduction is performed by *guided filtering* [38], since it provides high-quality edge-preserving smoothing at real-time rates. The guide for filtering was the input image itself. Contrast enhancement is achieved by *unsharp masking* [39,40], the essence of which is to sharpen the input image by adding its estimated high-frequency components, i.e., its *unsharp mask* to itself:(1)R=P+λU,
where R is the output image, P is the input image, λ is the degree of enhancement, and U is the unsharp mask, which is obtained by subtracting the low-pass filtered version of the input image from itself, i.e., $U=P-P_{\mathrm{filtered}}$. In our solution, the low-pass filtered version could simply be set to the output of the guided filter. In this case, however, the color channels would be sharpened separately, causing the hues of the pixels to change, which results in color noise. To preserve the hues of the pixels, we apply unsharp masking only to the luminance channel, i.e., our enhancement formula is
(2)$$R_{i}=P_{i}\left(1+\lambda \left(1-\frac{\mathrm{lum}(P_{\mathrm{filtered}}i)}{\mathrm{lum}(P_{i})}\right)\right),$$
where $\mathrm{lum}(P_{i})$ is the luminance of pixel $P_{i}$. For simple calculations, we approximate the luminance as the average of the red, green, and blue intensities.

In the last stage of pre-processing, the enhanced image is cropped to remove any margins and then resized to 160×160 pixels, which was found to be the optimal input size for the neural network. Afterwards, data are standardized to have a mean of zero and a standard deviation of one, which improves the stability and accuracy of the predictions.

Figure 3 displays the output of pre-processing using different guided filter and unsharp masking parameters. In our system, both the guided filter radius *r* and the unsharp masking strength λ are adjustable real numbers between 0 and 5 (0: no effect, 5: strong effect, i.e., strong blurring in the case of filtering or strong sharpening in the case of unsharp masking).

#### 2.2.2. Segmentation

The pre-processed image is fed into a deep neural network that predicts the class of each pixel. Eight models were compared for the segmentation task at hand, which are listed in Section 2.1. All models used the MobileNetV3-Large network [41] as an encoder, which is an efficient, lightweight CNN designed to perform well with limited computing resources or strict time constraints. Since we aim to provide a real-time tool to assist bladder tumor resection, the small size and low latency of the model are key factors.

A total of 7 classes are distinguished in the image:Normal tissue;Bladder cancer;BCG cystitis;TURBT scar;Ureteric orifice;Air bubble;Image artifact (instrument, scope, overlay text, etc.).

The output of the neural network is a seven-element vector for each pixel of the input image, where the *i*th element corresponds to the confidence level of the network in that the given pixel belongs to the *i*th class. Pixels are classified in the category with the highest confidence level.

#### 2.2.3. Post-Processing

In the post-processing phase, the polygon outlines of the detected lesions are extracted from the segmentation map and drawn on the original cystoscopic image (Figure 1). The color of the polygon indicates the class of the lesion; e.g., bladder cancer is marked by cyan, while artifacts are marked by green.

To obtain the outlines of the detected lesions, the boundaries of the homogeneous regions of the segmentation map must be determined. A pixel is at a boundary if one of its neighbors belongs to a different class than itself. However, merely collecting the boundary pixels can result in a noisy and complicated outline. To simplify the boundary, pixels where the curvature would be too high are removed. The easiest way to achieve this is to count how many pixels belong to another class in the eight-neighborhood of the given pixel, and if we obtain a number greater than four, the pixel is removed. Note that this procedure reduces the area of the polygon; therefore, the number of its repeated applications should be limited.

The polygon outlines of the regions of a given class are formed according to the following iterative algorithm. First, an arbitrary pixel is selected from the boundaries of that class, i.e., we find a pixel that belongs to the given class but at least one of its neighbors does not. In each subsequent step, we search for the next boundary pixel until we return to the starting point, thus closing the loop. During the search, we first examine the immediate neighborhood of the currently selected pixel, beginning with the direction taken in the previous step. If we find a boundary pixel of the same class, the pixel is added to the polygon as a vertex, and is also removed from the segmentation map so that it is not re-selected later. If no other boundary pixels are found in the immediate neighborhood of the current one, which happens if there is a loop in the outline, we start examining the pixels that are farther away, i.e., at a distance of two pixels. For performance reasons, the search distance is maximized to five pixels. If there are no other boundary pixels in the five-pixel-wide neighborhood, the polygon is considered to be noise and discarded.

The presented algorithm produces a single, closed outline. In cases where multiple regions belong to the given class, the procedure should be repeated until all boundary pixels are removed.

Prediction errors can result in unrealistically small polygons that need to be identified and discarded. We calculate the area of the polygon of vertices ${\vec{r}}_{1},\ldots ,{\vec{r}}_{n}$ as
$$A=\frac{1}{2}\left|\sum_{i=1}^{n}{\vec{r}}_{i}\times {\vec{r}}_{i+1}\right|,$$
where ${\vec{r}}_{n+1}={\vec{r}}_{1}$. If the resulting area *A* is small, e.g., smaller than 0.25% of the total area of the image (64 pixels in the case of a 160×160 pixel input), then the polygon is discarded.

### 2.3. Network Training

To improve segmentation accuracy, transfer learning [42] is used. That is, the neural networks are pre-trained on the ImageNet database [10] before being trained on cystoscopic images. This way, the networks learn a general model of the visual world that can facilitate the interpretation of cystoscopic images.

There are relatively few samples in our dataset for three of the seven classes (ureteric orifice, TURBT scar, BCG cystitis), as a result of which the networks have difficulty learning them. To address this class imbalance, we use *focal loss* [43], which is a modification of the standard cross-entropy loss that focuses training on hard samples. The focal loss dynamically scales the cross-entropy loss by a scaling factor that decays to zero as confidence in the correct class increases. This scaling factor automatically downweights the contribution of easy samples during training and focuses the model on samples it finds harder to classify.

To present the formal definition of the focal loss, let us start from the cross-entropy (CE) loss for binary classification (For simplicity, we only discuss binary classification, but extension to the multiclass case is straightforward [43]):$$\mathrm{CE}\left(p,y\right)=\left\{\begin{array}{cc} -\log(p) & \mathrm{if}\;y=1,\\ -\log(1-p) & \mathrm{otherwise}, \end{array}\right.$$
where y∈{±1} denotes the ground truth class and p∈[0,1] is the probability estimated for the class with label y=1. To simplify the notation, let us introduce
$$p_{t}=\left\{\begin{array}{cc} p & \mathrm{if}\;y=1,\\ 1-p & \mathrm{otherwise}, \end{array}\right.$$
in which case $\mathrm{CE}\left(p,y\right)=\mathrm{CE}\left(p_{t}\right)=-\log\left(p_{t}\right)$.

Focal loss adds a modulating factor ${\left(1-p_{t}\right)}^{\gamma }$ to the cross-entropy loss, where γ≥0 is a tunable *focusing* parameter. That is, focal loss is defined as
(3)$$\mathrm{FL}\left(p_{t}\right)=-{\left(1-p_{t}\right)}^{\gamma }\log\left(p_{t}\right).$$

It can be observed that as the confidence in the correct class increases, i.e., $p_{t}\to 1$, the modulating factor converges to 0, and the loss for well-classified samples is downweighted. In contrast, if a sample is misclassified and $p_{t}$ is small, the factor is approximately equal to 1, and thus, the loss is unaffected.

The focal loss has an α-balanced version, which was reported to give slightly better results than the original form [43]. In this version, a hyperparameter α∈[0,1] is introduced as a weighting factor for the positive class, while (1−α) is used to weight the negative class. Defining $\alpha_{t}$ analogously to $p_{t}$, the α-balanced version of the focal loss is calculated as
(4)$$\mathrm{FL}\left(p_{t}\right)=-\alpha_{t}{\left(1-p_{t}\right)}^{\gamma }\log\left(p_{t}\right).$$

In our solution, the α for a given class is inversely proportional to the relative frequency of occurrence of the class in the training set. That is, the more pixels are labeled by that class in the training set, the lower value its α weight will have.

### 2.4. Measurement Setup

#### 2.4.1. Data

We used a private dataset consisting of 2578 cystoscopy images manually annotated by medical professionals. Data were split into train, validation, and test sets with a ratio of 90:5:5, i.e., 2320 of the images were put in the train set, 129 were put in the validation set, and 129 were put in the test set. The number of image samples was increased by augmenting the dataset with the two pre-processing techniques presented in Section 2.2.1: noise reduction by guided filtering and contrast enhancement by unsharp masking. In total, each image was saved in four different versions (Figure 4):*Original*: no enhancement;*Filtered*: only guided filtering (r=5);*Sharpened*: only unsharp masking (λ=5);*Enhanced*: both guided filter and unsharp masking.

This way, the number of image samples was quadrupled, so the total training set consisted of 9280 images, the validation set consisted of 516 images, and the test set consisted of 516 images.

The training set was also augmented on-the-fly using the following transformations:Random changes in intensity (hue, brightness, contrast, saturation, value, gamma);Gaussian noise;Gaussian blur;Sharpening by the imgaug [44] library;Horizontal and vertical flipping;Rotating;Dropping out random 8×8 pixel squares.

These augmentations were different in every epoch; therefore, the network was never trained on the exact same image twice.

#### 2.4.2. Implementation

The neural networks were implemented in Python, using the PyTorch framework [45] and the Segmentation Models library [46]. Training was performed for 1000 epochs by the Adam optimizer [47] with a learning rate of $7\cdot 10^{-5}$ and a batch size of 64. The loss function was the α-balanced focal loss described in Section 2.3, with γ=2 and the α of each class pre-calculated from the statistics of the dataset. The parameters of the segmentation models were not changed from the default settings of the Segmentation Models library. All calculations were run on an NVIDIA GeForce RTX 2080 SUPER GPU [48].

#### 2.4.3. Evaluation Metrics


*Per-pixel Dice coefficient*


The segmentation maps generated by the neural networks were evaluated using the Dice similarity coefficient [49]. Given the set *T* of pixels belonging to a given class *c*, and the set *P* of pixels predicted by the network to belong to class *c*, the Dice coefficient is calculated as
(5)$$\mathrm{DICE}=\frac{2\;|T\cap P|}{\left|T|+|P\right|},$$
where |·| denotes the cardinality of a set and ∩ denotes the intersection of two sets. This calculation is performed for each class, and the results are averaged. The higher the Dice coefficient, the closer the predicted segmentation map is to the real one, i.e., the better the localization accuracy of the lesions is.


*Per-image F-score*


The main hazards are producing a false negative or false positive classification for the input image. For example, if an image is declared cancer-free while in truth it shows a cancerous region, the patient may go undiagnosed and not receive the necessary treatment. Conversely, if an image is declared to show cancer when it is in truth either fully normal or has only benign features, the patient may receive unnecessary treatment, which is expensive and could pose health risks. Therefore, we group classes as

*Malignant*: bladder cancer;*Benign*: TURBT scar, BCG cystitis;*Healthy*: normal, ureteric orifice, air bubble, image artifact;and counted images in the following categories:


*True Positive Malignant*: the classification indicates a malignant region, and the image indeed has such a region;*True Negative Malignant*: the classification indicates no malignant region, and the image indeed has no such region;*False Positive Malignant*: the classification indicates a malignant region, but the image has no such region;*False Negative Malignant*: the classification indicates no malignant region, but the image has such a region;*True Positive Benign*: the classification indicates a benign feature (but no malignant region), and the image indeed shows a benign feature (and no malignant region);*True Negative Benign*: the classification indicates healthy tissue (neither malignant nor benign feature), and the image indeed shows healthy tissue;*False Positive Benign*: the classification indicates a benign feature (but no malignant region), but the image shows healthy tissue;*False Negative Benign*: the classification indicates healthy tissue (neither malignant nor benign feature), but the image has a benign feature.


From these categories, the following metrics can be derived to describe the per-image classification accuracy:*Precision*: the fraction of patients with a true lesion (malignant or benign) among those who were identified to have the lesion:
$$precision=\frac{TP}{TP+FP}.$$*Recall (= Sensitivity)*: the fraction of patients identified with a lesion (malignant or benign) among those who truly have a lesion:
$$recall=\frac{TP}{TP+FN}.$$*F-score*: a measure combining the precision and recall of the classification system. It is defined as the harmonic mean of the model’s precision and recall:
$$\begin{array}{cc} Fscore & =\frac{2}{\frac{1}{precision}+\frac{1}{recall}}=\frac{TP}{TP+\frac{1}{2}\left(FP+FN\right)}. \end{array}$$

The primary objective is to maximize the F-score for the malignant category, as a misclassification of bladder cancer can have serious consequences. In contrast, benign features usually do not require treatment; at most, they are monitored for a certain period of time. Therefore, the classification accuracy of benign features is only secondary. The per-pixel localization accuracy (i.e., the Dice coefficient) serves as a tertiary aspect.

## 3. Results

### 3.1. Per-Pixel Dice Coefficient

The Dice coefficient was evaluated for each image in the test set. Table 1 presents the mean Dice scores averaged on the test images. The results show that all models were able to achieve a 91–92% mean Dice score, which is very promising, especially given that our dataset was relatively poorly balanced. The highest mean Dice score (92.91%) was obtained by the PAN [27] architecture.

Figure 5 displays the segmentation maps generated by the different models for a grade 2 papillary bladder cancer. All models were able to detect the tumor, although the resulting shapes were generally simpler and more ellipsoidal than the zigzag-edged ground truth mask. The highest Dice score for this image (98.67%) was achieved by the U-Net model.

### 3.2. Per-Image F-Score

Table 2 and Table 3 present the per-image classification accuracy of the trained models evaluated on the test set. The highest F-score for the malignant category was achieved by the PAN model, which was closely followed by the U-Net and the UNet++ networks. For the benign category, the highest F-score was achieved by DeepLabv3, which was closely followed by PAN.

F-scores were also evaluated separately on the image subsets of the four different pre-processed versions (original, filtered, sharpened, enhanced) in order to investigate which pre-processing method improves the classification accuracy the most. This measurement was performed using the PAN model. Results are shown in Table 4 and Table 5. It can be observed that the enhanced pre-processing method leads to the best lesion detection ability for both the malignant and benign categories.

### 3.3. Speed

The CNN should predict segmentations fast enough to be applied in real time on video streams during the transurethral resection of bladder tumors. To see if this is accomplished, we have measured the time required to calculate a single prediction, averaged from 100,000 runs on a NVIDIA GeForce RTX 2080 SUPER GPU [48]. The frame loading time or the time needed to fetch the image from the camera is not included, since it depends on the actual format, disk, and camera used. Measurement results are presented in Table 6.

The highest prediction speed (6.72 ms) was achieved by the DeepLabv3+ model. Note that even the slowest model (MA-Net) took less than 9 ms to make a prediction.

The pre- and post-processing of images requires some additional time. Guided filtering takes on average 2 ms and unsharp masking takes another 0.02 ms to perform on an NVIDIA GeForce GTX 960M GPU [50]. The extraction of polygon contours from the segmentation map takes an average of 0.06 ms, but the exact speed depends on the number of detected lesions and the complexity of their contours. Note that even with pre- and post-processing, a frame rate of 93–114 fps is attainable.

### 3.4. Model Size

The neural network is preferred to have a small size so that it does not occupy much storage space, it is fast to load (requires low bandwidth and has low memory latency), and it does not pose a problem to keep the model in the RAM or cache. Table 7 summarizes the sizes of the trained models. It can be observed that the model sizes range from 7 to 69 MB, which should pose no problem to store and utilize even on a clinical infrastructure with limited computing resources. Furthermore, the PAN model that achieved the highest prediction performance occupies only 7.93 MB, which is practically the smallest size of all.

A summary of the evaluation results is displayed in Table 8.

## 4. Discussion

The segmentation of cystoscopic images is challenging for several reasons. Firstly, the images are often blurry and have low contrast due to difficult shooting conditions. Secondly, the appearance of bladder lesions shows a large variance, even within a single class. Thirdly, for CNNs that generally work with fixed receptive fields, the presence of objects at multiple scales causes difficulties in classification [27,28]. This cannot be avoided in cystoscopy, as lesions have different physical sizes and, in addition, their dimensions in image space depend on the current position and orientation of the cystoscope.

Image quality can be improved with appropriate pre-processing, while the problem of high variance within classes can be mitigated by providing a sufficiently large and varied training dataset. However, the effective recognition of objects at multiple scales must be achieved by the neural network architecture.

U-Net, UNet++, and LinkNet do not give a direct solution for multi-scale object segmentation. Nevertheless, information about objects at different scales is implicitly encoded at the different levels of the U-shaped architecture. Skip connections also help deliver this information to the decoder where segmentation takes place.

U-Net/UNet++ and LinkNet differ in the way their decoder blocks combine the features coming from the skip connection and the previous decoder block. In U-Net/UNet++, these two feature maps are concatenated, while in LinkNet, they are summed to reduce the number of parameters. Our results show that it is not always advantageous to save on the parameters: both the per-image classification accuracy and the per-pixel localization accuracy of LinkNet were significantly lower than that of the U-Net models. This may indicate that semantic information can be lost during the summation of feature maps.

A technique widely used in segmentation models to achieve multi-scale object recognition is the feature pyramid method. Image features are extracted in a pyramidal fashion at incremental scales, and different levels are used to predict masks at different scales. In general, the lower levels of the pyramid carry semantically less accurate, but spatially higher resolution information, whereas the higher levels of the pyramid carry semantically more accurate but spatially lower resolution information. In order to combine the advantages of the two types, it is common to merge the features of different pyramid levels.

Among the neural networks we investigated, FPN, PAN, DeepLabv3, and DeepLabv3+ use feature pyramids to provide multi-scale object detection capability. However, their methods of creating the feature pyramid differ.

DeepLabv3 and DeepLabv3+ use dilated convolutions with an increasing dilation rate to generate the feature pyramid. MA-Net also uses dilated convolutions to obtain multi-scale context. However, this type of convolution has the inherent problem that when the dilation is large, the convolutional kernel becomes too sparse to cover any local information [51]. This may impair the local consistency of feature maps [27].

FPN uses simple convolutions and spatial upscaling to generate the feature pyramid, which has a total of four levels. The final segmentation map is then obtained as the sum of all pyramid level predictions, which are weighted by 1/4, 1/8, 1/16, and 1/32, respectively. However, FPN did not perform well in our evaluations compared to the other models. We argue that this is due to the fixed weights assigned to the pyramid levels. Depending on how close the cystoscopic camera is to the examined lesions, different object scales, i.e., different pyramid levels may provide the most important semantic information.

PAN uses a pre-trained ResNet-101 to create the feature pyramid and then applies a Feature Pyramid Attention (FPA) module to fuse the features of different levels. FPA has a U-shaped structure that integrates attention-inspired information from different scales step-by-step in a precise and efficient way. In addition, PAN uses global context information to guide segmentation via Global Attention Upsampling modules. This increases the receptive field of the model and enhances the consistency of pixel-wise classification.

We believe that PAN was able to achieve the best accuracy in cystoscopic image segmentation because it efficiently combines the feature pyramid method, the attention mechanism, and also makes use of global context information. However, these operations entail a significant computational cost, making PAN one of our slowest networks. In contrast, the dilated convolutions used by the DeepLabv3 models are very fast to compute.

The 92.91% Dice score achieved by our PAN model indicates better tumor localization ability than reported in previous studies. As listed in Section 1, the attention mechanism-based U-Net of Zhang et al. [19] achieved a Dice score of 82.5%, the mask region-based CNN of Yoo et al. [6] obtained a score of 74.7%, while the CystoNet of Shkolyar et al. [4] did not report the Dice score result, but their per-frame sensitivity of 90.9% is lower than the 94.79% of our PAN model. This improvement in accuracy may be the result of more appropriate pre-processing of the input image, a better choice of neural network architecture, or a difference in the dataset.

In contrast to previous studies, we distinguish several types of lesions in addition to bladder cancer. Results show that the neural network can detect and localize them all with good accuracy. This can not only reduce the chance of any abnormalities going unnoticed but may also help urologists differentiate between benign and malignant lesions.

We also observed that the proposed pre-processing techniques (noise reduction, contrast enhancement) improved the accuracy of the subsequent segmentation step compared to the case when the raw input image is fed into the neural network. In the future, we hope to investigate more image enhancement algorithms to see if further improvements can be made. Moreover, with the rapid development of segmentation neural networks, we believe that architectures more efficient than the current ones may appear within a few years, which should be examined for cystoscopy.

We consider the greatest advantages of our solution to be the advanced and configurable image enhancement framework as well as the possibility of segmenting multiple types of bladder lesions. Disadvantages include the scaling of the input images to a fixed size of 160 × 160 pixels as well as the fact that it is not possible to know in advance what settings should be used for the pre-processing of the individual image frames in order to achieve the best possible segmentation accuracy.

## 5. Conclusions

In this paper, we examined a deep learning-based solution for the automatic segmentation of cystoscopy images. As a pre-processing, noise was reduced by guided filtering, and contrast enhancement was performed by unsharp masking. Then, the image was fed into a CNN that predicted a high-accuracy segmentation map, marking bladder cancer, two types of non-cancerous lesions, and four types of healthy tissue. This map was post-processed to find the boundary polygons and overlay the input image with their outlines, assisting doctors in cancer diagnosis and treatment. We examined eight popular neural network architectures for the task at hand, and we found that the PAN model achieves the best results with a mean Dice score of 92.91% and a malignant F-score of 91%. The small model size and the few millisecond prediction time of the neural network allow the solution to be applied in real time on video streams during the transurethral resection of bladder tumors.

## Figures and Tables

**Figure 1 diagnostics-12-02849-f001:**
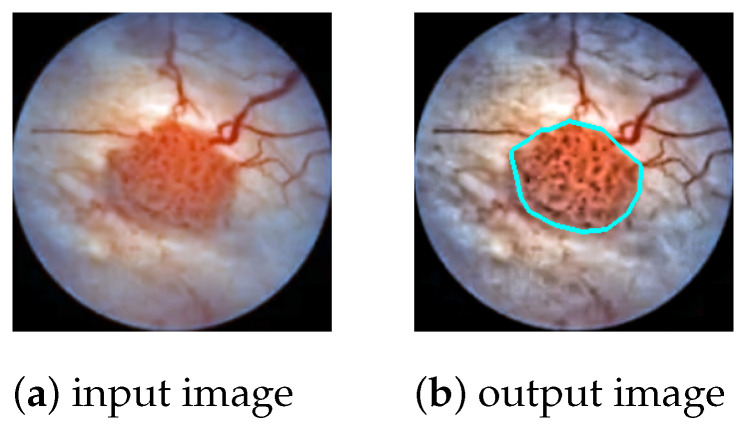
An example of a segmentation outline overlay. The input image shows a grade 2 bladder tumor that has been well localized by the DeepLabv3+ [29] neural network.

**Figure 2 diagnostics-12-02849-f002:**
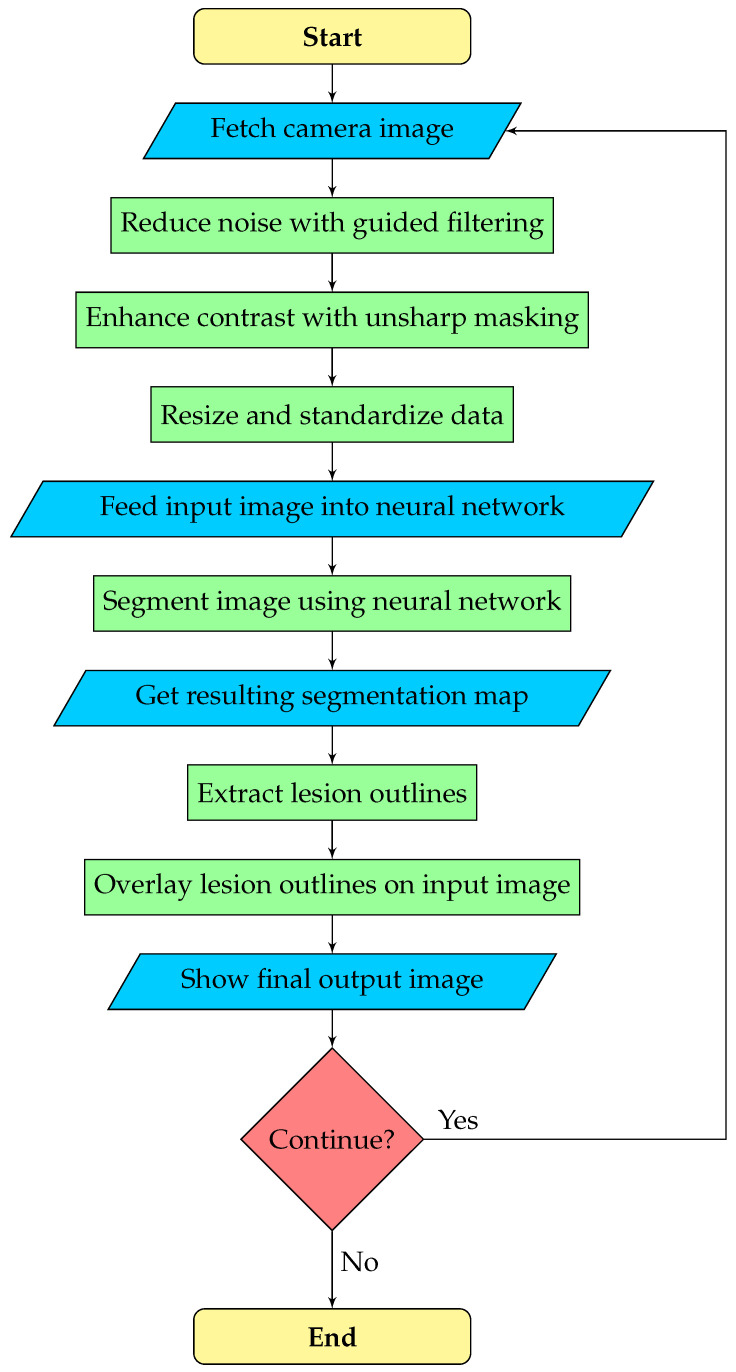
Flowchart of the cystoscopic image segmentation pipeline.

**Figure 3 diagnostics-12-02849-f003:**
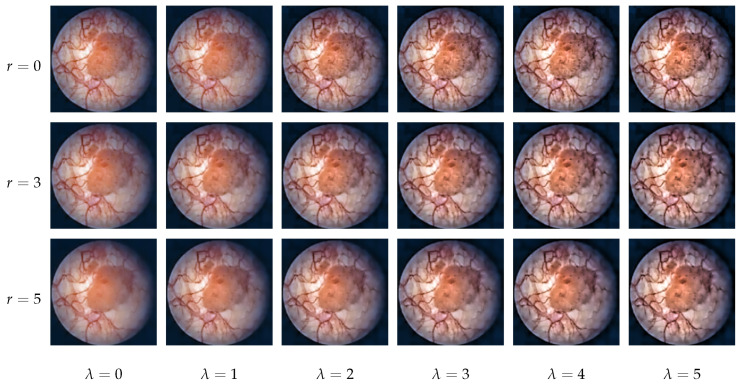
Pre-processing outputs using different guided filter radii *r* and unsharp masking strength λ values. The original input image is shown in the top left corner (r=0,λ=0).

**Figure 4 diagnostics-12-02849-f004:**
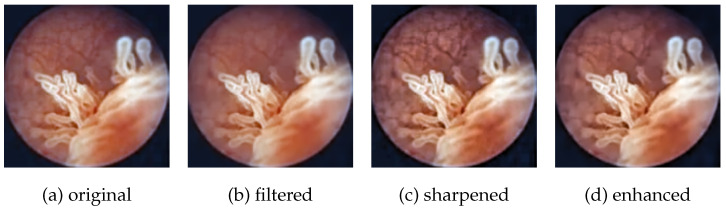
Four different pre-processing outputs of the same input image. In (**a**), the image was only cropped. In (**b**), guided filtering was applied to reduce noise, with radius r set to 5. In (**c**), unsharp masking was applied to sharpen the image, with strength λ set to 5. In (**d**), both guided filtering and unsharp masking were applied (r = 5, λ = 5).

**Figure 5 diagnostics-12-02849-f005:**
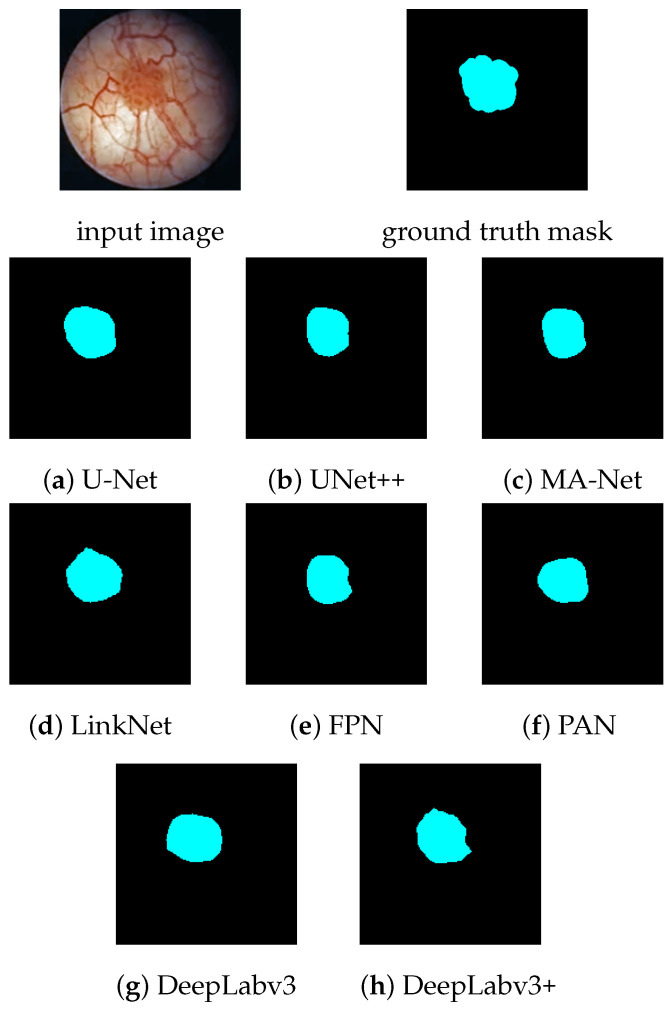
Segmentation maps predicted by the different models for an input image with a grade 2 papillary bladder tumor. Dice scores: (**a**) 98.67%, (**b**) 97.13%, (**c**) 97.34%, (**d**) 98.43%, (**e**) 97.34%, (**f**) 97.44%, (**g**) 97.96%, and (**h**) 98.59%.

**Table 1 diagnostics-12-02849-t001:** Mean Dice scores achieved by the different models on the test set.

**Model**	**Mean Dice Score [%]**
U-Net	92.54
UNet++	92.73
MA-Net	91.78
LinkNet	89.66
FPN	92.69
PAN	92.91
DeepLabv3	91.57
DeepLabv3+	92.47

**Table 2 diagnostics-12-02849-t002:** Per-image classification accuracy analysis for the malignant category, evaluated on the test dataset. The highest F-score was achieved by the PAN model.

**Model**	**TP**	**TN**	**FP**	**FN**	**Sensitivity**	**Precision**	**F-Score**
U-Net	187	290	34	5	97.3958	84.6154	90.5569
UNet++	187	291	33	5	97.3958	85.0000	90.7767
MA-Net	178	282	42	14	92.7083	80.9091	86.4078
LinkNet	181	281	43	11	94.2708	80.8036	87.0192
FPN	172	295	29	20	89.95833	85.5721	87.5318
PAN	182	298	26	10	94.7917	87.5000	91.0000
DeepLabv3	182	289	35	10	94.7917	83.8710	88.9976
DeepLabv3+	180	294	30	12	93.7500	85.7143	89.5522

**Table 3 diagnostics-12-02849-t003:** Per-image classification accuracy analysis for the benign category, evaluated on the test dataset. The highest F-score was achieved by the DeepLabv3 model.

**Model**	**TP**	**TN**	**FP**	**FN**	**Sensitivity**	**Precision**	**F-Score**
U-Net	49	212	11	18	73.1343	81.6667	77.1654
UNet++	50	208	10	23	68.4932	83.3333	75.1880
MA-Net	36	208	6	32	52.9412	85.7143	65.4545
LinkNet	52	193	20	16	76.4706	72.2222	74.2857
FPN	46	209	8	22	67.6471	85.1852	75.4098
PAN	54	216	8	20	72.9730	87.0968	79.4118
DeepLabv3	54	209	12	14	79.4118	81.8182	80.5970
DeepLabv3+	47	216	9	22	68.1159	83.9286	75.2000

**Table 4 diagnostics-12-02849-t004:** Malignant category F-scores evaluated all the test images and on the image subsets of the four different pre-processed versions (original, filtered, sharpened, enhanced), using the PAN model. The highest F-score was achieved on the enhanced data subset.

**Model**	**TP**	**TN**	**FP**	**FN**	**Sensitivity**	**Precision**	**F-Score**
all	184	302	22	8	95.8333	89.3204	92.4623
original	46	75	6	2	95.8333	88.4615	92.0000
filtered	46	75	6	2	95.8333	88.4615	92.0000
sharpened	46	75	6	2	95.8333	88.4615	92.0000
enhanced	46	77	4	2	95.8333	92.0000	93.8776

**Table 5 diagnostics-12-02849-t005:** Benign category F-scores evaluated all the test images and on the image subsets of the four different pre-processed versions (original, filtered, sharpened, enhanced), using the PAN model. The highest F-score was achieved on the enhanced data subset.

**Model**	**TP**	**TN**	**FP**	**FN**	**Sensitivity**	**Precision**	**F-Score**
all	64	213	12	13	83.1169	83.2105	83.6601
original	16	53	3	3	84.2105	84.2105	84.2105
filtered	15	53	3	4	78.9474	83.3333	81.0811
sharpened	16	53	3	3	84.2105	84.2105	84.2105
enhanced	17	54	3	3	85.0000	85.0000	85.0000

**Table 6 diagnostics-12-02849-t006:** The mean prediction time of the different models on a single image, calculated from 100,000 runs on a NVIDIA GeForce RTX 2080 SUPER GPU [48].

**Model**	**Mean Runtime (ms)**
U-Net	7.11
UNet++	8.04
MA-Net	8.74
LinkNet	6.96
FPN	7.36
PAN	7.46
DeepLabv3	6.76
DeepLabv3+	6.73

**Table 7 diagnostics-12-02849-t007:** The sizes of the trained neural networks.

**Model**	**Model Size (MB)**
U-Net	19.1
UNet++	20.0
MA-Net	69.4
LinkNet	10.0
FPN	14.3
PAN	7.93
DeepLabv3	31.2
DeepLabv3+	12.7

**Table 8 diagnostics-12-02849-t008:** Summary of the evaluation results that measure the means of the per-pixel Dice score (%), the per-image F-score (%), the prediction time (ms) and the model size (MB). In terms of overall prediction performance, the PAN model proved to be the best.

**Model**	**Dice Score**	**Malignant F-Score**	**Prediction Time**	**Model Size**
U-Net	92.54	90.5569	7.1133	19.1
UNet++	92.73	90.7767	8.0443	20.0
MA-Net	91.78	86.4078	8.7378	69.4
LinkNet	89.66	87.0192	6.9627	10.0
FPN	92.69	87.5318	7.3593	14.3
PAN	92.91	91.0000	7.4643	7.93
DeepLabv3	91.57	88.9976	6.7604	31.2
DeepLabv3+	92.47	89.5522	6.7270	12.7

## Data Availability

Not applicable.

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
