# Peer review of "A Comparative Study of Deep Neural Networks for Real-Time Semantic Segmentation during the Transurethral Resection of Bladder Tumors"

_diagnostics, 2022, doi:10.3390/diagnostics12112849_

Round 1

Reviewer 1 Report

  1. The authors are suggested to add a strong motivation behind taking up this problem.
  2. Add a brief paragraph at the end of the introduction section that describes the paper's organization.
  3. A few details need to be included in equation (2).
  4. In figure 3, what is known as r, and what is the boundary of r and lamda? 
  5. On page 4, the paragraph before 2.3 needs to be clarified; please reframe the sentence.
  6. Correct a few typos (e.g., Seach)
  7. Add a block diagram or flow chart of the proposed method.
  8. Please add the demographic information to the data.
  9. The discussion must be different from the conclusion. 
  10. Add pros and cons of the proposed method.

Author Response

Dear Reviewer,

Thanks for the rigorous and constructive reviews. We have taken them into account and re-written the paper accordingly. Please find the explanation how we addressed the concerns below.

Sincerely Yours,

The authors

  1. The authors are suggested to add a strong motivation behind taking up this problem.

Answer: The Introduction section has been extended and now has a more prominent motivational narrative. The main points are:

  • neural networks showed promising results in previous cystoscopy studies, but there is still room for improvement in terms of bladder tumor localization accuracy (previous Dice scores were low),
  • classical image segmentation architectures achieve great precision in other applications, so they should be examined for cystoscopy.
  1. Add a brief paragraph at the end of the introduction section that describes the paper's organization.

Answer: The paragraph presenting the organization of the paper has been added to the end of the Introduction section, on page 2.

  1. A few details need to be included in equation (2).

Answer: An explanation has been provided for Equation 2. We clarified what we mean by luminance: for simplicity, we simply calculate the average of the red, green, and blue intensities of the pixel.

  1. In figure 3, what is known as r, and what is the boundary of r and lamda?

Answer: The meaning of r is described in the caption as well as at the end of Section 2.2.1: it is the radius parameter of guided filtering. The boundaries of r and lambda have been added to the end of Section 2.2.1, on page 5.

  1. On page 4, the paragraph before 2.3 needs to be clarified; please reframe the sentence.

Answer: The sentence has been clarified and now includes the precise threshold we used for discarding polygons based on their areas.

  1. Correct a few typos (e.g., Seach)

Answer: We corrected

  • “seach” to “search” on page 6
  • “one of the first study” to “one of the first studies” on page 1
  • “classificaton" to “classification” on page 2
  • “Shihoya” to “Shkolyar” on page 2
  • “msec” to “ms” in the speed analysis on page 14 (to be consistent)
  • “low size” to “small size” on page 14
  • “non-cancerious” to “non-cancerous” on page 16
  • “U-Net++” to “UNet++” at each occurrence (the latter is the form used in the publication of the model)
  • “DeepLabV3” to “DeepLabv3” at each occurrence (the latter is the form used in the publication of the model)

and made some grammatical corrections in addition to typo correction.

  1. Add a block diagram or flow chart of the proposed method.

Answer: A flowchart has been added as Figure 2, displayed on page 6.

  1. Please add the demographic information to the data.

Answer: The hospitals providing the dataset removed all personal data, so no demographic information is available.

  1. The discussion must be different from the conclusion.

Answer: A separate Conclusions section (Section 5) has been added to the end of the paper, and the Discussion section (Section 4) has also been expanded.

  1. Add pros and cons of the proposed method.

Answer: Advantages and disadvantages of our method have been added to the end of the Discussion section, on page 16.

Reviewer 2 Report

The authors compared different deep learning-based solutions to detect bladder cancer and other lesions on cystoscopic images with high precision. Mainly, the method consists of three phases (i) a pre-processing phase to reduce noise, a segmentation phase where the CNN models are used, and a post-processing phase in which the predicted segmentation map is overlaid on the cystoscopic image to highlight lesions. The authors evaluated eight standard models (U-Net, U-Net++, MA-Net, LinkNet, FPN, PAN, DeepLabV3, DeepLabV3+) wrt accuracy and speed. The PAN model showed the best segmentation map precision, while the highest speed was obtained by DeepLabV3+. Here are my comments:

1. The authors claim that they have proposed a novel deep learning-based segmentation system for white-light cystoscopic images. However, the authors used some standard models. So novelty is not there.

2. The authors claim that they have proposed a 3-step pipeline consisting of pre-processing (image enhancement), segmentation via the neural network, and post-processing (overlay of segmentation results). These steps are normal processes to apply any DNN models. Nothing is new here.

3. The authors did not discuss the models used.

4. The title is confusing. The title should be changed. The title can be “A comparative study of….”.

5. The authors need to compare the results with other research works. Very few recent works is mentioned in the reference.  

6. Minimum discussion of the implementation of the models

Author Response

Dear Reviewer,

Thanks for the rigorous and constructive reviews. We have taken them into account and re-written the paper accordingly. Please find the explanation how we addressed the concerns below.

Sincerely Yours,

The authors

  1. The authors claim that they have proposed a novel deep learning-based segmentation system for white-light cystoscopic images. However, the authors used some standard models. So novelty is not there.
  2. The authors claim that they have proposed a 3-step pipeline consisting of pre-processing (image enhancement), segmentation via the neural network, and post-processing (overlay of segmentation results). These steps are normal processes to apply any DNN models. Nothing is new here.

Answers 1&2: We have removed these claims and shifted the focus of the paper to the comparison of neural network models for cystoscopic image segmentation. The Title, the Abstract, the Introduction, the Materials and Methods, the Discussion, and the Conclusion have been rewritten accordingly.

  1. The authors did not discuss the models used.

Answer: We have added a new section to Materials and Methods that describe the neural network architectures examined in the paper. It is Section 2.1, “The deep neural networks to be compared” on page 3-4.

  1. The title is confusing. The title should be changed. The title can be “A comparative study of….”.

Answer: The title has been changed to “A Comparative Study of Deep Neural Networks for Real-Time Semantic Segmentation During the Transurethral Resection of Bladder Tumors”.

  1. The authors need to compare the results with other research works. Very few recent works is mentioned in the reference.

Answer: More reference works have been added to the Introduction section. Furthermore, a comparison of our results and the results of the most relevant reference works has been added to the Discussion section.

  1. Minimum discussion of the implementation of the models

Answer: Section 2.4.2. has been supplemented with more details of our implementation. An even more thorough description can be found in the documentation of the Segmentation Models library that we used and cited.

Round 2

Reviewer 2 Report

The authors now shifted the claim of novelty. Now present a comparative analysis of eight state-of-the-art DNNs  for semantic segmentation of white-light cystoscopy images. Seems like a good comparative analysis. 

Please highlight the best result achieved in the abstract.

It would be better to discuss why  PAN model produces the best F-score for segmentation map precision, and the highest speed is obtained by DeepLabv3+. The analysis will give benefit other researchers in other image analysis research.

Author Response

Dear Reviewers,

Thanks for the review.

In the revised version, we have extended the abstract to highlight the best results. Additionally, we significantly expanded the Discussion section and now explain why the PAN architecture performs better in the given task than other compared networks.

Best regards,

The Authors